# Localization and Expression of Renin–Angiotensin System Receptors in Lung from Transplant Patients: A Case-Control Study

**DOI:** 10.3390/biomedicines13092312

**Published:** 2025-09-21

**Authors:** Andresa Thomé Silveira, Lucas Sagrillo Fagundes, Juliane Flor, Isabel Amaral Martins, Laura Bastos Otero, Laura Tibola Marques da Silva, Lorenzo Santana Maciel, Sarah Eller, Giuliano Rizzotto Guimarães, Fabíola Adelia Perin, Márcia Rosângela Wink, Katya Rigatto

**Affiliations:** 1Laboratório de Fisiologia Translacional, Universidade Federal de Ciências da Saúde de Porto Alegre (UFCSPA), Porto Alegre 91501-970, RS, Brazil; andresasilveira@ufcspa.edu.br (A.T.S.);; 2Programa de Pós-Graduação em Ciências da Saúde, Universidade Federal de Ciências da Saúde de Porto Alegre (UFCSPA), Porto Alegre 91501-970, RS, Brazil; 3Laboratório de Biologia Celular, Universidade Federal de Ciências da Saúde de Porto Alegre (UFCSPA), Porto Alegre 91501-970, RS, Brazil; 4Departamento de Patologia, Universidade Federal de Ciências da Saúde de Porto Alegre (UFCSPA), Porto Alegre 91501-970, RS, Brazil; 5Serviço de Cirurgia Torácica, E de Transplante da Irmandade da Santa Casa de Misericórdia de Porto Alegre (ISCMPA), Porto Alegre 91501-970, RS, Brazil

**Keywords:** idiopathic pulmonary fibrosis, renin–angiotensin system, lung transplantation, spirometry

## Abstract

**Objective**: We aimed to assess the expression and localization of renin-angiotensin system (RAS) receptors in lung tissue and the plasma concentration of related peptides in IPF patients. **Materials and Methods**: This case–control study involved 19 patients from southern Brazil undergoing lung resection or transplantation. Plasma levels of Angiotensin I, II, A, 1-7, Alamandine were measured via liquid chromatography–tandem mass spectrometry. Lung tissue expression and localization of angiotensin type 1 (AT1), Mas, and Mas-related G-protein-coupled receptor D (MrgD) receptors were evaluated using Western blot and immunohistochemistry. Clinical data and the 6-min walk test were analyzed to correlate receptor expression with lung function and oxygen dependence. **Results**: IPF patients showed reduced forced vital capacity (FVC) at 49 ± 13% and forced expiratory volume (FEV1) at 51 ± 14%, with a 60% increase in oxygen dependence. Plasma peptide concentrations were similar between the groups, except for Angiotensin I, which was significantly higher in the control group. In IPF lungs, AT1 and Mas receptors were expressed 2.31 and 2.13 times more, respectively, while MrgD expression was lower. Mas receptors were mostly found in bronchiole areas, whereas MrgD was predominant in the lung parenchyma. **Conclusions**: This study indicates that the RAS operates independently within tissue, in addition to its systemic functions, highlighting distinct differences between tissue and plasma RAS activities. The distinct roles of MrgD and Mas receptors in lung structure and function could be pivotal for new therapies, potentially leading to more effective IPF treatments.

## 1. Introduction

Idiopathic pulmonary fibrosis (IPF) is the most prevalent interstitial lung disease, characterized by a progressive and chronic course that remains not fully understood [1,2]. It involves the remodeling of the interstitium in distal airways and alveolar spaces, predominantly affecting the posterior basilar regions of the lung, typically beginning in a subpleural distribution [3].The pathogenesis of IPF is complex, marked by repetitive scarring of the alveolar tissue [4], and progresses slowly but is ultimately fatal, with an average survival of 2 to 4 years after diagnosis [5]. It significantly impairs respiratory capacity and is a global concern due to its clinical severity [6]. IPF is more prevalent among males in their sixth decade of life, largely attributed to the aging population [4,7]. In Brazil, approximately 16,000 IPF cases [6] are reported, with an expected rise partly attributable to the increased use of electronic cigarettes.

As the disease progresses, respiratory capacity is significantly impaired, with a persistent dry cough and dyspnea as the most common symptoms. Patients may also experience acute exacerbations that necessitate oxygen supplementation [7]. Currently, antifibrotic therapy is recommended in cases of IPF; however, the drugs Nintedanib and Pirfenidone delay disease progression but do not reverse fibrosis. Due to significant side effects and limited efficacy [8,9], it is crucial to study the pathophysiology of the disease for evaluating alternative therapeutic approaches, potentially leading to more effective treatments.

In Brazil, lung transplantation is almost the only option available for these individuals. In 2024, a total of 93 lung transplants were performed in the country, with 35 taking place in the state of Rio Grande do Sul [10]. However, data from the Brazilian Association of Organ Transplants in 2025 indicates that 348 people are still waiting for this lifesaving procedure [11].

Given the limited treatment options and the high demand for transplants in Brazil, there is an urgent need to explore other therapeutic avenues that could reduce patients’ reliance on transplant waiting lists. One promising area of interest is the renin–angiotensin system (RAS), which is well known for its role in promoting fibrosis through Angiotensin II (AngII) [12,13]. Attempts to treat fibrosis have focused on using AT1 receptor blockers and angiotensin-converting enzyme (ACE) inhibitors [9], which target AngII’s fibrotic pathways. Unfortunately, these treatments did not yield significant improvements in halting fibrosis progression, leading to a decline in research focus on the RAS. However, the discovery of new peptides within this system has reignited interest, highlighting the importance of maintaining a balance between the fibrotic axis driven by AngII and the antifibrotic pathways involving Angiotensin-(1-7) (Ang-(1-7)) and Alamandine (ALA) [14].

The understanding of the RAS has evolved significantly with the discovery of new peptides and enzymes, shifting from the classic view to a contemporary perspective. This includes Ang-(1-7), ALA, and ACE2, which play a crucial role in maintaining balance in the lungs, a key site for RAS activation [15]. Recent studies in patients with IPF demonstrated that this system has been evaluated in patients with IPF, revealing a probable imbalance in the RAS axes [16].

In the classic axis, the AT1 receptor primarily mediates the effects of AngII, which can lead to harmful outcomes such as inflammation and fibrosis when this pathway becomes dominant [17]. Conversely, other peptides and their receptors, such as Ang-1-7, which binds to the Mas receptor, exhibit vasodilatory, anti-fibrotic and anti-inflammatory effects. Additionally, the discovery of the peptide ALA, which binds to the MrgD receptor, show similar actions to Ang-(1-7) [18], providing protective effects on the lungs [19].

Due to the severity of IPF and the possible RAS imbalance involved, a deeper understanding of its pathophysiology may support the development of alternative treatments, either alone or in combination, that improve IPF management. In this scenario, this study aims to investigate the location and expression of the AT1, Mas, and MrgD receptors in the lung tissue and of Angiotensin I (Ang I), Ang II, Angiotensin A (Ang A), Ang-(1-7) and ALA peptides in the plasma of patients with and without IPF.

## 2. Materials and Methods

This article is part of a larger study approved by the Research Ethics Committee of the Universidade Federal de Ciências da Saúde de Porto Alegre and the Irmandade Santa Casa de Misericórdia de Porto Alegre (Number 69947517.2.0000.5345/69947517.2.3001.5335). All procedures followed the principles of the Declaration of Helsinki.

This case–control study involved 19 intraoperative patients undergoing surgical procedures, including lobectomy and/or pulmonary segmentectomy (control group) and lung transplantation (IPF group), at a renowned transplant hospital in southern Brazil. From May 2021 to May 2023, 1 cm^3^ of lung tissue and 4 mL of blood samples were collected in EDTA tubes with protease inhibitor P8340 (Sigma-Aldrich^®^, Darmstadt, Germany). Plasma was collected from the blood sample after refrigerated centrifugation. Lung tissue samples for the IPF group were collected from diseased lung tissue, while for the control group, they were collected from the most distal margin of the resected tissue, where there is the highest likelihood of representing normal, healthy characteristics. The collected plasma and tissues were frozen in nitrogen liquid and stored at −80 °C.

The inclusion criteria were adult IPF patients over the age of 18, of both sexes, who had undergone lung transplantation, and patients undergoing lung resection to treat bronchial carcinoma. Exclusion criteria included patients with clinical disorders that could compromise their participation or performance in the study, as well as those diagnosed with systemic arterial hypertension and/or taking ACE inhibitors, beta-blockers, or angiotensin receptor blockers.

Demographic and clinical data such as age, race, gender, weight, height, body mass index, smoking status, oxygen use, spirometry, lung scintigraphy, systolic pulmonary artery pressure (SPAP), and the 6-min walk test (6MWT) were collected from secondary data in the patients’ medical records. The data collected was stored in an electronic database and treated confidentially.

### 2.1. Western Blot Analysis of Pulmonary Ras Receptor Expression

Thawed lung tissue samples (−80 °C) were maceraded in liquid nitrogen, homogenized using RIPA Lysis Buffer (10x, Merck Millipore^®^, Darmstadt, Germany), protease inhibitor P8340 (Sigma-Aldrich^®^), and ultrapure water, followed by centrifugation at 15,000× *g* for 20 min at 4 °C. The protein concentration was de termined using the DC^TM^ protein assay kit (Bio-Rad^®^, Hercules, CA, USA)

To ensure consistent protein amounts, 75 μg of protein were subjected to one-dimensional sodium dodecyl sulfate-polyacrylamide gel electrophoresis (SDS-PAGE) using a batch system with a 10% gel. Samples were loaded in volumes ranging from 20 µL to 30 µL and allowed to migrate for 90 min.

In a cooled Bio-Rad transfer unit, the separated proteins were then transferred by electrophoresis to membranes using buffer containing 20 mM Tris, 150 mM glycine, 20% (*v*/*v*) methanol, and 0.02% (*w*/*v*) SDS (pH = 8.2). Membranes were incubated for 1.5 h in a blocking solution (5% (*w*/*v*) skimmed milk in 0.1% (*w*/*v*) Tris-Tween 20 buffer to prevent non-specific binding. The membranes were processed by immunodetection using the following primary antibodies: angiotensin II AT1 receptor polyclonal antibody (Enzo^®^, Farmingdale, NY, USA), 1:1000; MGPRF polyclonal antibody (Thermo Fisher Scientific^®^, Waltham, MA, USA), 1:1250; and Mas polyclonal antibody (Novus Biologicals^®^, Centennial, CO, USA), 1:1000.

The samples were then incubated with goat anti-rabbit IgG secondary antibodies conjugated to peroxidase (HRP; Sigma-Aldrich^®^) at a 1:5000, which were subsequently reacted with a chemiluminescent substrate. Band intensity was determined by quantitative densitometry analysis using Image Lab Software 6.0 (Bio-Rad^®^). The results were normalized by densitometry of the total protein in these samples, and protein levels were expressed as a ratio of the density of a specific band to the total protein stained using the Pierce Reversible Protein Stain Kit (Thermo Fisher Scientific^®^, Waltham, MA, USA).

### 2.2. Immunohistochemical Mapping of RAS Receptors in Lung Tissue

The specimens were fixed in 10% buffered formalin solution for 48 h to ensure optimal preservation, after which they were dehydrated, clarified and embedded in paraffin. The 4-micrometer histological sections were placed on salinized slides, and the immunohistochemical assay was carried out. The slides were heated in an oven at 75 °C for 30 min to aid in deparaffinized, followed by xylene treatment and gradual rehydrated in a graded series of ethanol followed by distilled water. Antigen retrieval for anti-Mas1 Polyclonal antibody (ref. NBP1-60091, Novus Biologicals^®^, Centennial, CO, USA) and anti-MRGPRF Polyclonal antibody (ref. PA5-110966, Thermo Fisher Scientific^®^, Waltham, MA, USA) was performed in a water bath for 40 min at 95 °C in 20 mM Tris/EDTA buffer (pH 9.0).

Endogenous peroxidase activity was blocked with 3% hydrogen peroxide in methanol, protected from light, and applied three times for 10 min each. Protein blocking was achieved with 1% BSA in PBS for 1 h. The sections were incubated overnight at 2–8 °C with the primary antibodies anti-Mas (dilution 1:200) and anti- MRGPRF (dilution 1:200). Subsequently, the HRP-conjugated polymer (Envision + Dual Link-HRP system ref: K4061, Dako, CA, USA) was applied and incubated for 30 min at room temperature. Diaminobenzidine (DAB) substrate (Liquid DAB + Substrate Chromogen System, ref: K3468, Dako, CA, USA) was used to visualize the reactions.

Negative controls, essential for validating specificity, were processed using the same protocol, omitting the primary antibody and replacing it with BSA solution. The slides were contrasted with Harris hematoxylin for 20 s. The sections were dehydrated in absolute ethanol, clarified in xylene, and mounted with synthetic mounting medium. Images were captured using a Leica DM6 microscope and analyzed qualitatively to verify receptor immunostaining in lung tissue.

### 2.3. Determination of Ang I, Ang II, Ang A, Alamandine, and Ang-(1-7) by Liquid Chromatography–Tandem Mass Spectrometry (LC-MS/MS)

An aliquot of 950 μL acetone was added to 50 μL of plasma samples and shaken for 60 s. After centrifugation at 9000× *g* for 6 min, the supernatant was collected, dried under nitrogen, and resuspended in 50 μL of acetonitrile immediately before analysis.

The analytical system included Nexera-i LC-2040C coupled with an LCMS-8045 triple quadrupole mass spectrometer (Shimadzu, Kyoto, Japan). Electrospray ionization parameters in positive mode were set as follows: capillary voltage at 4500 V; desolvation line temperature at 250 °C; heating block temperature at 400 °C; drying gas flow at 10 L/min; and nebulizing gas flow at 3 L/min. Multiple reaction monitoring (MRM) was conducted using the following fragmentation patterns. Ang I: m/z 649.10→110.10; 649.10→269.10; 649.10→426.25. Ang II: m/z 523.80→263.20; 523.80→70.20; 523.80→110.15. Ang A: m/z 501.80→70.10; 501.80→263.10; 501.80→110.05. ALA: m/z 428.45→110.10; 428.45→211.15; 428.45→11.10. Ang-(1-7): m/z 450.30→110.10; 450.30→70.10; 450.30→392.65.

Chromatographic separation was performed using an Acquity UPLC^®^ C18 column (2.1 × 50 mm, 1.7 μm particle size) (Waters Corporation^®^, Wexford, Ireland) in gradient elution mode at a flow rate of 0.3 mL/min. The mobile phase consisted of water (solvent A) and acetonitrile (solvent B), both fortified with 0.1% formic acid, programmed as follows: 0–0.5 min, 2% B; 0.5–3.0 min, 2–100% B; 3.0–3.5 min, 100% B; 3.5–3.8 min, 100–2% B; 3.8–8 min, 2% B. The column oven was maintained at 50 °C. Data were processed using LabSolutions software (Shimadzu, Kyoto, Japan).

### 2.4. Statistical Analysis

Data analysis was conducted using GraphPad Prism 7. Student’s *t*-test was used for statistical comparisons, and Pearson’s correlation test assessed the association between variables. All results are presented as mean ± standard deviation. The level of significance was set at *p* < 0.05.

## 3. Results

Clinical data and lung tissue from 19 patients, including those with and without IPF, were assessed. The mean age was 68 years (±13) for the IPF group and 59 years (±9) for the control group (*p* = 0.092). The IPF group included 4 males (44%), while the control group comprised 7 men (70%) (*p* = 0.369). A significant proportion of patients were identified as white, 9 (100%) in the IPF group and 8 (80%) in the control group (*p* = 0.156). Control group patients exhibited a normal body mass index (BMI) of 23.11 (± 4.57), whereas IPF patients were overweight, with a BMI of 25.81 (± 1.73) (*p* = 0.100). A minority of patients were active smokers: 11% in the control group and 20% in the IPF group (*p* = 0.999; Table 1).

Table 2 provides a comprehensive clinical characterization of patients, offering details on continuous oxygen use, spirometric data, lung scintigraphy, echocardiography, and the 6MWT. The analysis of these results revealed that only patients in the IPF group required continuous oxygen therapy, with a prevalence of 60% (*p* = 0.0108). Spirometric data indicated that all measured parameters were higher in the control group compared to the IPF group. The latter exhibited a moderate obstructive ventilatory disorder, with predicted forced vital capacity (FVC) at 48.72% ± 12.69 and predicted forced expiratory volume in the first second (FEV1) at 51.29% ± 14.58 (both *p* < 0.0001).

The echocardiogram results showed a right ventricular ejection fraction of 67.10 ± 7.34%, reflecting cardiac function, and a SPAP of 53 ± 29.89 mmHg, which is typical in IPF. Lung scintigraphy patterns carbon monoxide diffusing capacity (DLCO) had a mean value of 32.83 ± 13.43. In the 6MWT assessment, the distance covered by the IPF group (393.67 ± 73.71 m) was significantly lower than the estimated distance (556.67 ± 64.97 m).

Furthermore, this analysis revealed a significant increase in heart rate (HR) (from 81.44 ± 12.47 to 120.6 ± 13.32; *p* < 0.0001) post-test compared to pre-test results, respiratory rate (RR) (from 21.22 ± 6.77 to 37.22 ± 13.49; *p* = 0.005), BORG index (from 0.83 ± 1.17 to 3.94 ± 2.37; *p* = 0.002), and a decrease in oxygen saturation (SPO_2_) (from 96.11 ± 1.90 to 79.56 ± 6.04; *p* < 0.0001).

Figure 1 illustrates the protein expression of the RAS receptors in lung tissue, highlighting the protein bands and total protein levels for each receptor using the Western blot technique. The IPF group showed significantly higher protein expression of AT1 and Mas receptors, with increases of 2.13-fold and 2.31-fold, respectively, compared to the control group. Conversely, the MrgD receptor exhibited greater expression in the lung tissue of the control group, with a 1.83-fold increase compared to the IPF group.

In Figure 2 the identification of the Mas and MrgD receptors in lung tissue is presented, specifically within the pulmonary bronchiole and lung parenchyma. This identification was achieved using the immunohistochemistry technique. However, AT1 receptor was not detectable in the positive control when employing this method.

Table 3 provides the plasma characterization of the patients’ RAS, offering details on the peptides Ang I, Ang II, Ang A, Ang-(1-7), and ALA. Analysis of these results revealed that all measured parameters were higher in the control group compared to the IPF group, with only Ang I present statistical significance.

Correlations between clinical data and RAS peptides/receptors were performed, showing weak to moderate correlations, both positive and negative, without statistical significance. Only two correlations were statistically significant: the pre-BORG index with the Mas receptor, r = 0.679 (*p* < 0.05), and the ejection fraction with the AT1 receptor, r = −0.645 (*p* < 0.05).

## 4. Discussion

The main findings of this study are as follows: (1) the MrgD receptor is less expressed in the lung parenchyma of patients with IPF; (2) in contrast, the Mas and AT1 receptors are more highly expressed in the lungs of IPF patients; and (3) the Mas receptor is predominantly localized in the pulmonary bronchioles. Regarding the circulating components of the RAS, the plasma concentration of Ang I was significantly higher in the control group, while no significant differences were observed for Ang II, Ang-(1-7), or ALA (Figure 3). These findings support the concept of tissue-specific activation of the RAS, independent of the systemic circulation. Furthermore, our results align with previous findings in animal models [20,21] and add new evidence by demonstrating Mas receptor localization in the human lung, reinforcing its potential relevance in pulmonary fibrosis (PF) pathophysiology.

Our findings align with established epidemiological patterns of IPF prevalence [7,15,16,22]. Patients also presented higher body weight and BMI compared to controls, consistent with findings by Alakhras [23] and Souza et al. [24]. Although BMI values were within the range considered appropriate for older adults in Brazil (>22 to <27 kg/m^2^) [25], excess weight could potentially contribute to worsened respiratory function in these patients.

Disease severity was evidenced by altered spirometry, reduced 6MWD, and the need for continuous oxygen therapy. Most patients exhibited more severe clinical conditions than those described in other studies [16,22], particularly among those undergoing lung transplantation, as previously reported [23]. In addition, the marked reduction in FEV_1_ (%) and abnormalities observed on lung scintigraphy further underscore the advanced impairment of pulmonary function in our PF cohort. Consistently, previous studies have demonstrated that patients requiring continuous oxygen therapy often exhibit significant alterations in lung imaging, underscoring the urgent need for more effective therapeutic interventions.

Our 6MWT findings aligned with those of Sipriani et al. [16], and post-test changes in HR, RR, SpO_2_, and Borg scale scores confirmed impaired exercise tolerance, consistent with Morales-Blanhir et al. [26]. This limitation contributes to a higher risk of cardiovascular complications, as also described in the literature [27]. Moreover, patients exhibited severely reduced DLCO (≈33%) and FVC values, indicating significant functional impairment, in contrast to the expected DLCO range of 80–120% in healthy individuals [28,29].

Severe lung function impairment in IPF patients is confirmed by significant reductions in FEV1 and abnormal lung scintigraphy findings. Consistent with these findings, studies indicate that patients requiring continuous oxygen therapy often present significant lung scan abnormalities [30].

Although plasma levels of angiotensin peptides—except for Ang I—did not differ significantly between IPF patients and controls, lung tissue RAS receptor expression showed a distinct local activation pattern. Specifically, the increased expression of AT1 and Mas receptors in patients with IPF indicates an upregulation of tissue-specific signaling pathways, which may suggest a localized compensatory mechanism, independent of systemic peptide concentrations.

This is consistent with prior evidence in mice showing that in fibrotic lungs, AT1 receptor expression is enhanced, promoting Transforming Growth Factor beta mediated profibrotic signaling, fibroblast proliferation, and extracellular matrix deposition [31,32], even when circulating Ang II levels remain unchanged [32]. Ang II has a very short half-life in the circulation—less than 60 s [33] —which supports the hypothesis that it may be generated near its site of action, functioning as part of a localized RAS in various tissues [34]. Molecular studies indicate that AT1 activation induces both rapid cellular responses and long-term genetic changes [35]. Such findings underscore the importance of considering local receptor dynamics when interpreting the role of RAS in disease.

Furthermore, our findings align with the study by Raupp and colleagues [15], which also reported an increase in the counter-regulatory axis involving the AT1 in the lungs of IPF patients [15]. Known for mediating vasoconstriction and promoting fibrosis, the increased presence of AT1 receptor, along with reduced MrgD receptor expression in the parenchyma, could contribute to airway obstruction or reduced lung compliance.

Although RAS blockers have shown effects on lung tissue in patients [36] and animal models [37], our study highlights a divergence between local (organ-specific) and systemic RAS activities, which may explain why systemic RAS blockers have not proven clinically effective in treating IPF [38,39]. This concept is supported by studies demonstrating that the RAS operates both systemically and locally, with receptor expression patterns in the pulmonary vasculature and parenchyma responding to the tissue microenvironment [40]. This suggests that there may be underlying interactions or mechanisms between the tissue-specific and plasma RAS components that remain elusive, potentially impacting the overall therapeutic outcomes.

On the other hand, the upregulation of the Mas receptor in IPF may reflect an attempt to counteract fibrotic remodeling via the Ang-(1-7)/Mas axis [41], whereas the higher expression of MrgD receptor in control lungs—associated with Alamandine’s antifibrotic and anti-inflammatory actions—suggests a loss of this protective signaling in disease. Collectively, these findings indicate a pathological shift toward profibrotic dominance in the local RAS profile of IPF lungs, despite relatively stable systemic hormone levels.

Sipriani et al. [16] reported ALA plasma concentrations that were four times lower in patients with IPF. This is likely to reflect the fact that the patients in that study were in an earlier stage of fibrosis, where disease progression was clearly exacerbating due to the deficiency of ALA a peptide known for its anti-inflammatory and antifibrotic effects. It seems that the RAS becomes increasingly unbalanced as the disease advances, potentially contributing to the worsening of fibrotic conditions. The observed reduction in hormones as reported by Sipriani [16], combined with our findings of decreased MrgD receptor levels, indicates that this axis is significantly impaired in all stages of IPF in patients.

The distinct receptors localization found indicate specific pathophysiological roles upon stimulation. It is likely that MrgD receptors play a role in modulating gas exchange processes and influencing fibrotic responses. Conversely, the Mas receptors, located in the bronchioles, probably play a role in counteracting the airway obstruction. These roles underscore the complex interplay of these receptors in lung pathophysiology.

This rationale is confirmed in asthma [42], where Mas receptors, predominantly located in the bronchioles and the bronchi of animals [20,43], may influence airway resistance and ventilation dynamics. Understanding the precise localization of receptors in humans is crucial, as it suggests that therapies targeting specific receptors could be meticulously tailored to address pulmonary functions and pathological processes. This awareness allows for more precise interventions, potentially enhancing treatment efficacy and minimizing side effects.

While our study provides valuable insights, it is important to consider certain limitations that could help guide future research. Expanding the sample size and diversity could enhance the applicability of our findings across broader populations. Additionally, longitudinal studies might offer a deeper understanding into the causal relationships between receptor localization and disease progression. Exploring a wider array of RAS receptors and pathways could further elucidate the complex mechanisms at play in IPF.

## 5. Conclusions

This study underscores the critical importance of receptor localization within the lung tissue for understanding the pathophysiology of IPF and developing targeted therapies. The differential expression and localization of RAS receptors point to a complex interplay of local tissue dynamics that drive disease progression. Currently, there are no therapies that integrate considerations of local receptor environments. However, in the future, such insights highlight the importance of therapeutic strategies that consider these nuances to effectively modulate pathological processes. Moreover, recognizing the distinct roles of various receptors can inform the design of tailored therapies that optimize efficacy while minimizing adverse effects.

Overall, our results indicate the need for future treatment strategies for IPF that explore novel local components of the RAS. Our findings suggest the development of new therapeutic agents, which could offer patients better outcomes and an improved quality of life.

## Figures and Tables

**Figure 1 biomedicines-13-02312-f001:**
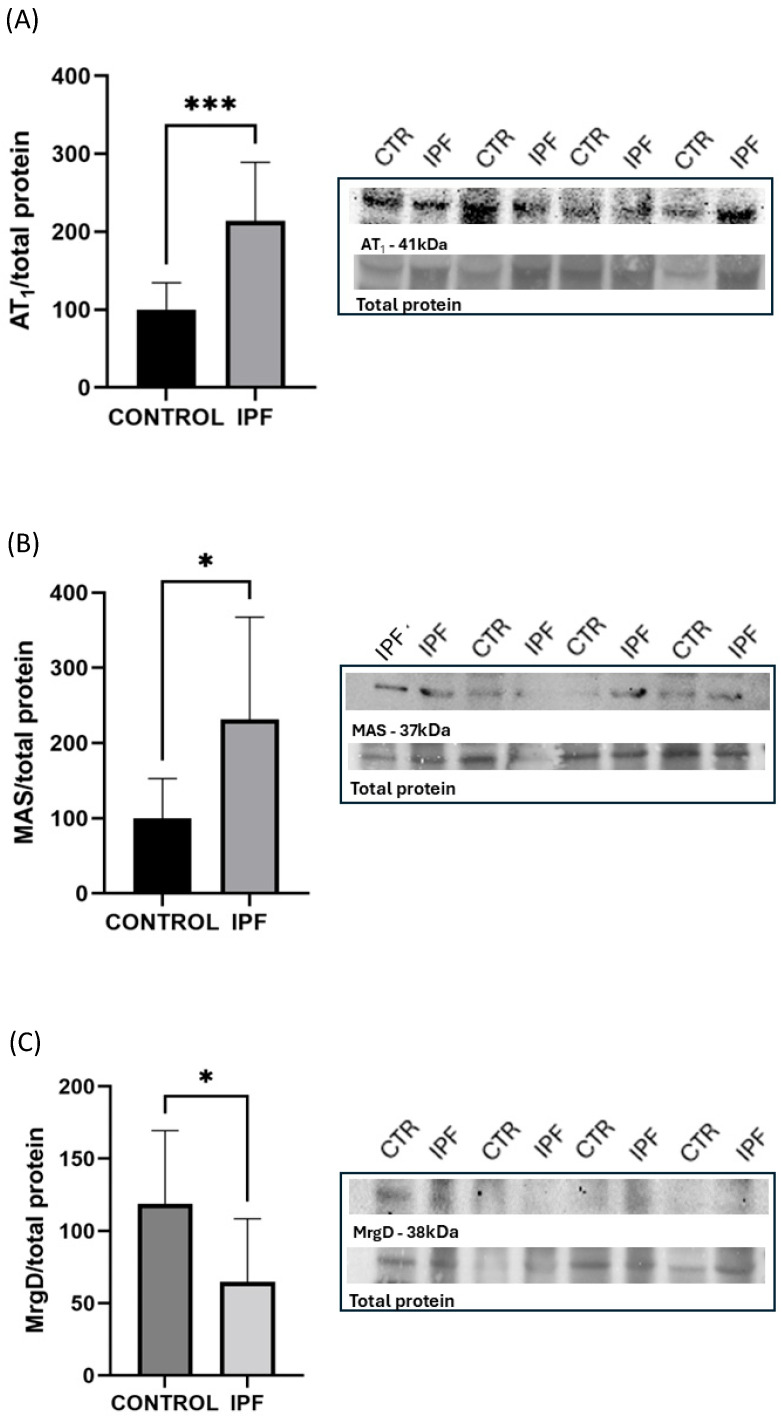
Total protein expression in lung homogenates of AT1, Mas, MrgD receptors was detected by immunoblotting using specific polyclonal antibodies. The molecular relative mass (kDa) is indicated to the right of the blot. Bar graphs represent the relative optical density of AT1: Angiotensin II Type 1 Receptor (**A**), Mas: Mas Receptor (**B**) and MrgD: Mas-Related G Protein-Coupled Receptor D (**C**) with total protein. Data presented are mean ± SD (*n*= 9 control group and *n* = 10 FPI group); were analyzed using *t*-Test (* *p* < 0.05; *** *p* < 0.001).

**Figure 2 biomedicines-13-02312-f002:**
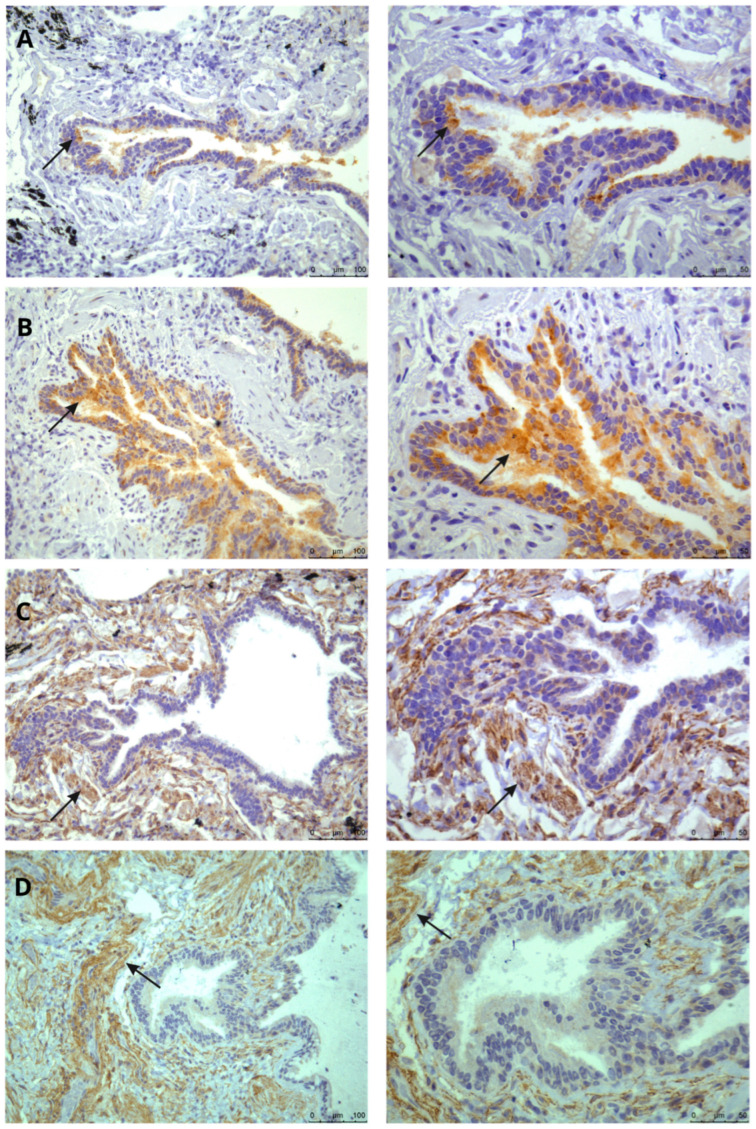
Photomicrograph for localization of Mas and MrgD receptors. The positive immunohistochemical reaction is identified by brown color, observed by microscopy at 20× magnification and 40× magnification. In (**A**), control group and (**B**) FPI group, presence of the Mas receptor (Mas Polyclonal Antibody (Novus Biologicals^®^, Centennial, CO, USA.—1:200) in the bronchiole. In (**C**), control group and (**D**) FPI group, presence of the MrgD receptor (MRGPRF Polyclonal Antibody (ThermoFisher Scientific^®^, Waltham, MA, USA.—1:200), in the lung parenchyma.

**Figure 3 biomedicines-13-02312-f003:**
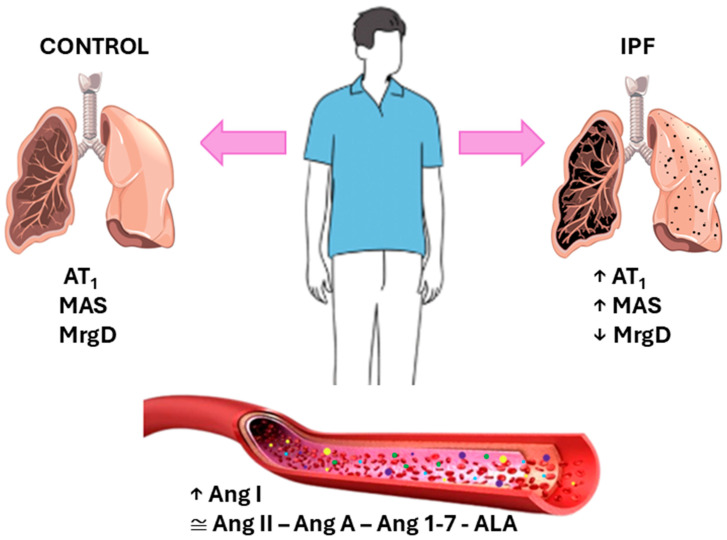
Involvement of the renin–angiotensin system (RAS) in idiopathic pulmonary fibrosis (IPF). The image of the fibrotic lung tissue shows an imbalance in the RAS receptors—AT1, Mas, and MrgD—compared to control tissues. The plasma levels of peptides such as Ang II, Ang A, Ang 1-7, and ALA are not significantly different between groups. AT1: Angiotensin II Type 1 Receptor; Mas: Mas Receptor; MrgD: Mas-Related G Protein-Coupled Receptor D; Ang II: Angiotensin II; Ang A: Angiotensin A; Ang 1-7: Angiotensin 1-7; ALA: Alamandine.

**Table 1 biomedicines-13-02312-t001:** Sample characterization.

Variables	Control *(n* = 9)	IPF (*n* = 10)	*p*
Age (years)	68 ± 13	59 ± 9	0.092
Male	4	7	0.369
Caucasian	9	8	0.156
Weight (Kg)	61.77 ± 13.8	74.20 ± 7.98	0.026 *
Height (m)	1.63 ± 7.53	1.69 ± 7.04	0.072
BMI (Kg/m^2^)	23.11 ± 4.57	25.81 ± 1.73	0.100
Previous smoking	6	6	
Never smoked	3	4	0.999

Values expressed as mean ± standard deviation; IPF, Idiopathic Pulmonary Fibrosis; Kg, Kilogram; m, meter; BMI, Body Mass Index. * *p* < 0.05.

**Table 2 biomedicines-13-02312-t002:** Patients clinical characterization.

Variables	Control (*n* = 9)	IPF (*n* = 10)	*p*
Oxygen use—n(%)	0	6 (60)	0.0108
Spirometry			
FVC (%)	97 ± 14.51	48.72 ± 12.69	0.0001 ****
FEV_1_ (%)	90.56 ± 11.33	51.29 ± 14.58	0.0001 ****
Echocardiogram			
EF(%)		67.10 ± 7.34	
SPAP (mmHg)		53 ± 29.89	
Lung scintigtaphy			
DLCO(%)		32.83 ± 13.43	
TC6M			
Distance traveled (m)		393.67 ± 73.71	
Estimated distance (m)		556.67 ± 64.97	
% predicted		70.88 ± 11.24	
HR—pre—post (bpm)		81.44 ± 12.47—120.6 ± 13.32	0.0001 ****
RR—pre—post (mpm)		21.22 ± 6.77—37.22 ± 13.49	0.005 **
BORG—pre—post		0.83 ± 1.17—3.94 ± 2.37	0.002 **
SPO2—pre—post		96.11 ± 1.90—79.56 ± 6.04	0.0001 ****

FVC, Forced vital capacity; FEV, Force expiratory volume in first second; EF, ejection fraction; SPAP, systolic pulmonary artery pressure. 6MWT, Six-Minute Walk Test; Pre and post: Pre- and Post-6MWT; HR, heart rate; RR, respiratory rate; BORG, BORG Scale; SPO_2_; oxygen saturation. Values expressed as mean ± standard deviation. ** *p* < 0.01; **** *p*< 0.0001.

**Table 3 biomedicines-13-02312-t003:** Renin–Angiotensin-System peptides plasma concentration (ng/mL).

RAS Peptides	Control (*n* = 9)	IPF (*n* = 10)	*p*
Angiotensin I	0.1703 ± 0.06185	0.1193 ± 0.02392	0.0449 *
Angiotensin II	0.06500 ± 0.01911	0.06088 ± 0.01027	0.5948
Angiotensin A	0.1733 ± 0.1543	0.1181 ± 0.05859	0.3572
Angiotensin 1-7	0.06088 ± 0.01241	0.05288 ± 0.009156	0.1645
Alamandine	0.1782 ± 0.04416	0.1479 ± 0.06302	0.2638

Values expressed as mean ± standard deviation. * *p* < 0.05.

## Data Availability

The raw data supporting the conclusion of this article will be made available by the authors upon request.

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
