# Peer review of "Localization and Expression of Renin–Angiotensin System Receptors in Lung from Transplant Patients: A Case-Control Study"

_biomedicines, 2025, doi:10.3390/biomedicines13092312_

Round 1
Reviewer 1 Report
Comments and Suggestions for Authors
The article "Localization and expression of Renin-Angiotensin system receptors in lung from transplant patients: a case-control study" by Silveira et al. focuses on the Renin-angiotensin system and its receptors found in the lungs of patients with idiopathic lung fibrosis and control patients, their characterization, and searches for the evidences, that RAS plays a significant role in the IPF development.
This is a well-written study, I have only several comments and suggestions:
- In the materials and methods section it should be noted - why the control patients were undergoing segmentectomy, and if it was cancer or other oncological diseases - it should be noted in this section, for clarity.
- In the Table 2 with clinical characterization of the patients - why the authors didnt measure systemic arterial pressure? To me it seems, that it is a suitable characteristic to indicate, especially considering the fact that the RAS plays a huge part in the regulation of AP, and some connections between the localization of RAS receptors and AP characterizations could be drawn.
- In the Figure 2 the scale bars should be added to the immunohistochemical images to increase the clarity of this figure.
- I usually dont note the stylistical choices, but in table 1 I could not help but notice the "White breed" variable in the third row. I suppose this is to note that the patients were white, as in Caucasian? I believe that "white breed" is not very suitable for the scientific publication concerning humans, and respectfully ask the authors to change this denomination to something more suitable, such as "Caucasian", or "White".
- In the discussion section I missed the discussion of the possible therapeutic application of the RAS modifying therapy strategies. If there is no such strategies at the present time, perhaps this should be specifically noted.
Author Response
Dear Referee, we want to express our sincere gratitude for the insightful reviews and constructive comments provided on our manuscript titled "Localization and Expression of Renin-Angiotensin System Receptors in Lung from Transplant Patients: A Case-Control Study." We are pleased to know that you appreciated the clarity of our writing and the significance of our findings.
Below, we address the reviewers' comments and suggestions in detail:
Comments and Suggestions for Authors
The article "Localization and expression of Renin-Angiotensin system receptors in lung from transplant patients: a case-control study" by Silveira et al. focuses on the Renin-angiotensin system and its receptors found in the lungs of patients with idiopathic lung fibrosis and control patients, their characterization, and sear ches for the evidences, that RAS plays a significant role in the IPF development.
This is a well-written study, I have only several comments and suggestions:
- In the materials and methods section it should be noted - why the control patients were undergoing segmentectomy, and if it was cancer or other oncological diseases - it should be noted in this section, for clarity.
RESPONSE: We appreciate this suggestion and have now included clarification regarding the reasons for the segmentectomy in the control patients in the Materials and Methods section (lines 113 and 114). We have also rewritten the exclusion criteria for clarity (lines 114 to 117).
- In the Table 2 with clinical characterization of the patients - why the authors didnt measure systemic arterial pressure? To me it seems, that it is a suitable characteristic to indicate, especially considering the fact that the RAS plays a huge part in the regulation of AP, and some connections between the localization of RAS receptors and AP characterizations could be drawn.
RESPONSE: Thank you for highlighting this important point. We have clarified in the methods section that patients did not have a diagnosis of hypertension, which aligns with our exclusion criteria stated in lines 114 and 117.
- In the Figure 2 the scale bars should be added to the immunohistochemical images to increase the clarity of this figure.
RESPONSE: We increased the size of Figure 2 to make the scale bars more visible, as they were previously too small for effective interpretation.
- I usually dont note the stylistical choices, but in table 1 I could not help but notice the "White breed" variable in the third row. I suppose this is to note that the patients were white, as in Caucasian? I believe that "white breed" is not very suitable for the scientific publication concerning humans, and respectfully ask the authors to change this denomination to something more suitable, such as "Caucasian", or "White".
RESPONSE: We want to thank you for your observation regarding the terminology. We fully agree and have updated the term from 'White breed' to 'Caucasian' in Table 1, as per your suggestion, to ensure the use of appropriate scientific language.
- In the discussion section I missed the discussion of the possible therapeutic application of the RAS modifying therapy strategies. If there is no such strategies at the present time, perhaps this should be specifically noted.
RESPONSE: We appreciate this insightful observation. We have revised the text to clarify that the new treatment strategies are ideas for the future and that alamandine should be considered in this context. This clarification can be found in lines 421-423. We do not address potential treatment strategies in this article, as discussing them would currently be considered speculative.

Reviewer 2 Report
Comments and Suggestions for Authors
Article
Localization and Expression of Renin-Angiotensin System Receptors in Lung from Transplant Patients: A Case-Control Study
Abstract of the work is well written, it clearly highlights the main and differential findings of RAS receptors in the lung of patients with IPF and that could become important for future therapies, in summary, it provides an overview and complete of the work.
The methodology is clear and well used. The results obtained in the study have important implications for the treatment of idiopathic pulmonary fibrosis (IPF): the anti-fibrotic and profibrotic pathways are highlighted, as well as the metabolic balance between them. The effect of tissue RAS, specific receptors, and localized therapies.
Mas receptor plays a role in pulmonary fibrosis, especially in IPF. According to the results of the study, Mas receptor is more expressed in the lungs of patients with IPF and is located predominantly in bronchioles. Receptor associated with anti-fibrotic pathway of RAS. In summary, the Mas receptor appears to have a protective role in pulmonary fibrosis, and its activation could be key to developing more effective treatments against IPF.
Analysis of figures and tables
Figures are clear and well-structured. Although in Figure 1, the bands are not clearly visible, especially in A (Western Blot analysis). The immunohistochemistry photos are very well achieved, and tables is clearly.
Discussion and conclusions
The discussion is well used, it is clear, and the authors integrate the results obtained (they use relevant citations).
Reference is made to the limitations of the study, which gives an adequate impression, as it is possibly the weakest point of the research. The conclusions are well founded.
Bibliography
On the other hand, the bibliography used is adequate, up-to-date, and uses relevant references to support the findings.
Therefore, the only thing I would add to this work is a scheme that facilitates the visualization of the most relevant findings and the questions left by this work.
Author Response
Open Review 2
Comments and Suggestions for Authors
Article
Localization and Expression of Renin-Angiotensin System Receptors in Lung from Transplant Patients: A Case-Control Study
Abstract of the work is well written, it clearly highlights the main and differential findings of RAS receptors in the lung of patients with IPF and that could become important for future therapies, in summary, it provides an overview and complete of the work.
The methodology is clear and well used. The results obtained in the study have important implications for the treatment of idiopathic pulmonary fibrosis (IPF): the anti-fibrotic and profibrotic pathways are highlighted, as well as the metabolic balance between them. The effect of tissue RAS, specific receptors, and localized therapies.
Mas receptor plays a role in pulmonary fibrosis, especially in IPF. According to the results of the study, Mas receptor is more expressed in the lungs of patients with IPF and is located predominantly in bronchioles. Receptor associated with anti-fibrotic pathway of RAS. In summary, the Mas receptor appears to have a protective role in pulmonary fibrosis, and its activation could be key to developing more effective treatments against IPF.
Analysis of figures and tables
Figures are clear and well-structured. Although in Figure 1, the bands are not clearly visible, especially in A (Western Blot analysis). The immunohistochemistry photos are very well achieved, and tables is clearly.
Discussion and conclusions
The discussion is well used, it is clear, and the authors integrate the results obtained (they use relevant citations).
Reference is made to the limitations of the study, which gives an adequate impression, as it is possibly the weakest point of the research. The conclusions are well founded.
Bibliography
On the other hand, the bibliography used is adequate, up-to-date, and uses relevant references to support the findings.
Therefore, the only thing I would add to this work is a scheme that facilitates the visualization of the most relevant findings and the questions left by this work.
RESPONSE:
We want to express our sincere gratitude for your thoughtful review of our manuscript entitled "Localization and Expression of Renin-Angiotensin System Receptors in the Lungs of Transplant Patients: A Case-Control Study." We are particularly grateful for your positive comments regarding the clarity of our writing and the importance of our findings.
We greatly appreciate your recognition of the well-structured abstract and the implications of our results for the future treatment of idiopathic pulmonary fibrosis. We are pleased to hear that you found the methodology clear and effective, as well as the discussion that appropriately integrated our results with relevant citations.
We also value your constructive feedback regarding the figures and your suggestion to include a diagram to visualize our main findings. Despite our graphical abstract, we have also included a figure after line 324.

Reviewer 3 Report
Comments and Suggestions for Authors
Overall, the manuscript is well structured and presents interesting and relevant findings regarding the role of the renin–angiotensin system in IPF. The combination of peptide quantification, receptor expression, and localization provides valuable insights into tissue-specific versus systemic RAS activity. However, before the manuscript can be considered for publication, several important methodological and interpretative issues need to be clarified. Addressing these points will strengthen the robustness of the conclusions and the overall impact of the study.
2-It is not clear why no housekeeping protein was used as a loading control. The manuscript mentions that results were normalized by total protein densitometry, but the method is not described in sufficient detail (e.g., staining procedure, quantification approach). Given that housekeeping proteins are the conventional normalization standard in Western blot, the lack of either a housekeeping control or a clear description of the total protein normalization method raises concerns about the robustness and reproducibility of the results.
3- While the immunohistochemistry protocol is described in detail, the manuscript does not provide sufficient information on how the images were analyzed. It is unclear whether the evaluation was qualitative or quantitative, what software or criteria were used for image analysis, and whether the assessment was blinded. These details are essential to ensure reproducibility, minimize bias, and allow proper interpretation of the findings.
4- The chromatographic conditions are generally well described. However, the injection volume and the total run time are not mentioned. These details are important to ensure reproducibility of the method.
5- The absence of AT1 receptor detection by immunohistochemistry requires clarification. It is not clear whether this negative result reflects a true lack of expression in the tissue, or whether it may be related to technical issues such as antibody sensitivity, specificity, or tissue processing. Given the central role of AT1 in the RAS pathway and its known relevance in fibrosis, it would be particularly important to validate its expression by immunohistochemistry and provide a more detailed explanation of this finding.
Author Response
Open Review 3
Comments and Suggestions for Authors
1-Overall, the manuscript is well structured and presents interesting and relevant findings regarding the role of the renin–angiotensin system in IPF. The combination of peptide quantification, receptor expression, and localization provides valuable insights into tissue-specific versus systemic RAS activity. However, before the manuscript can be considered for publication, several important methodological and interpretative issues need to be clarified. Addressing these points will strengthen the robustness of the conclusions and the overall impact of the study.
RESPONSE: We would like to extend our sincere gratitude for your thoughtful review of our manuscript titled "Localization and Expression of Renin-Angiotensin System Receptors in Lung from Transplant Patients: A Case-Control Study." We truly appreciate your concern for the quality of our work, as it is essential for the integrity of our research.
Your comments regarding the structure and relevance of our findings on the renin-angiotensin system (RAS) in idiopathic pulmonary fibrosis (IPF) are highly encouraging. We agree that addressing the methodological and interpretative issues you have highlighted will strengthen our conclusions and enhance the overall impact of the study.
2-It is not clear why no housekeeping protein was used as a loading control. The manuscript mentions that results were normalized by total protein densitometry, but the method is not described in sufficient detail (e.g., staining procedure, quantification approach). Given that housekeeping proteins are the conventional normalization standard in Western blot, the lack of either a housekeeping control or a clear description of the total protein normalization method raises concerns about the robustness and reproducibility of the results.
RESPONSE: We appreciate the value of using a housekeeping protein for normalization in Western blot analyses. However, it's important to highlight that the current gold standard in research does not necessarily involve comparing alpha-actin or GAPDH bands with those of the samples. Instead, the use of total protein measurement for western blot loading controls, known as total protein normalization (TPN), offers a solution to the inherent challenges of linearity in immunodetection for both target and control proteins. TPN accounts for the intensity of all proteins in the lane, as well as variations in sample loading, electrophoresis, and transfer processes. Notably, this method has been improved and can be viewed from lines 125 to 156.
For further reading, please refer to the following sources:
Bettencourt JW, McLaury AR, Limberg AK, et al. Total protein staining is superior to classical or tissue-specific protein staining for the standardization of protein biomarkers in heterogeneous tissue samples. Gene Reports. 2020 Jun;19:100641. doi: 10.1016/j.genrep.2020.100641
Taylor SC, Berkelman T, Yadav G, Hammond M. A defined methodology for reliable quantification of Western blot data. Mol Biotechnol. 2013 Nov;55(3):217-26. doi: 10.1007/s12033-013-9672-6.
3- While the immunohistochemistry protocol is described in detail, the manuscript does not provide sufficient information on how the images were analyzed. It is unclear whether the evaluation was qualitative or quantitative, what software or criteria were used for image analysis, and whether the assessment was blinded. These details are essential to ensure reproducibility, minimize bias, and allow proper interpretation of the findings.
RESPONSE: We appreciate your observation that more details about the type of immunohistochemistry assay were needed. To ensure clarity and reproducibility, we have included this information in the methods section on lines 187–188. Regarding receptor quantification, we utilized Western blotting to measure the receptors quantitatively. As for the localization within pulmonary tissue, this was assessed qualitatively, and therefore, we did not employ any software or specific criteria for image analysis to identify the location. See more details in the response to question number 5.
4- The chromatographic conditions are generally well described. However, the injection volume and the total run time are not mentioned. These details are important to ensure reproducibility of the method.
RESPONSE: We have included the injection volume and total run time in lines 127-151 of the revised manuscript, as these details are crucial to ensuring the reproducibility of our method.
5- The absence of AT1 receptor detection by immunohistochemistry requires clarification. It is not clear whether this negative result reflects a true lack of expression in the tissue, or whether it may be related to technical issues such as antibody sensitivity, specificity, or tissue processing. Given the central role of AT1 in the RAS pathway and its known relevance in fibrosis, it would be particularly important to validate its expression by immunohistochemistry and provide a more detailed explanation of this finding.
RESPONSE: Your question about the lack of detection of the AT1 receptor by immunohistochemistry is relevant. The presence of AT1 receptors in lung tissue was confirmed using Western blotting. However, our immunohistochemistry (IHC) results were inconclusive due to a critical limitation: the antigen-antibody did not stain in adipose tissue, which was used as our positive control. This failure undermines confidence in any IHC results for AT1 receptor in the lung tissue, as the lack of staining where expected suggests potential issues with the technique. We conducted multiple tests according to the ATLAS OF HUMAN PROTEINS, but without successful detection of AT1 in adipose tissue, we couldn't confirm its localization in lung tissue through IHC.

Round 2
Reviewer 1 Report
Comments and Suggestions for Authors
I thank the authors for addressing my concerns, and have no objections to the publication of the article in the present form.